# Advances in X-ray free electron laser (XFEL) diffraction data processing applied to the crystal structure of the synaptotagmin-1 / SNARE complex

Artem Y Lyubimov[1,2,3,4,5†], Monarin Uervirojnangkoorn[1,2,4,3,5†], Oliver B Zeldin[1,2,4,3,5], Qiangjun Zhou[1,2,4,3,5], Minglei Zhao[1,2,4,3,5], Aaron S Brewster[6], Tara Michels-Clark[6], James M Holton[6,7,8], Nicholas K Sauter[6], William I Weis[1,3,4*], Axel T Brunger[1,2,4,3,5*]

[1]Department of Molecular and Cellular Physiology, Stanford University, Stanford, United States; [2]Neurology and Neurological Science, Stanford University, Stanford, United States; [3]Structural Biology, Stanford University, Stanford, United States; [4]Photon Science, Stanford University, Stanford, United States; [5]Howard Hughes Medical Institute, Stanford University, Stanford, United States; [6]Molecular Biophysics and Integrated Bioimaging Division, Lawrence Berkeley National Laboratory, Berkeley, United States; [7]Stanford Synchrotron Radiation Lightsource, SLAC National Accelerator Laboratory, Menlo Park, United States; [8]Department of Biochemistry and Biophysics, University of California, San Francisco, San Francisco, United States

*For correspondence: bill.weis@stanford.edu (WIW); brunger@stanford.edu (ATB)

†These authors contributed equally to this work

**Abstract** X-ray free electron lasers (XFELs) reduce the effects of radiation damage on macromolecular diffraction data and thereby extend the limiting resolution. Previously, we adapted classical post-refinement techniques to XFEL diffraction data to produce accurate diffraction data sets from a limited number of diffraction images (*Uervirojnangkoorn et al., 2015*), and went on to use these techniques to obtain a complete data set from crystals of the synaptotagmin-1 / SNARE complex and to determine the structure at 3.5 Å resolution (*Zhou et al., 2015*). Here, we describe new advances in our methods and present a reprocessed XFEL data set of the synaptotagmin-1 / SNARE complex. The reprocessing produced small improvements in electron density maps and the refined atomic model. The maps also contained more information than those of a lower resolution (4.1 Å) synchrotron data set. Processing a set of simulated XFEL diffraction images revealed that our methods yield accurate data and atomic models.

## Introduction

X-ray free electron laser (XFEL) crystallography is an emerging technique for obtaining high-resolution diffraction data from macromolecular crystals (*Schlichting, 2015*). Diffraction data from an XFEL pulse lasting only tens of femtoseconds are largely free from X-ray induced radiation damage that might otherwise affect the success of crystallographic phasing and atomic model refinement. However, the crystal is effectively stationary during the XFEL pulse, which complicates determination of the crystal lattice model from the resulting zero-rotation or 'still' diffraction images. Furthermore, the XFEL pulse destroys or damages the illuminated crystal volume and thus allows only a single diffraction image to be collected. This effect is exacerbated by the variation in intensity and spectrum

of the incident XFEL beam produced by the self-amplified spontaneous emission (SASE) process (*Bonifacio et al., 1994*; *Emma et al., 2010*). Together, these features cause significant image-to-image variability in the diffraction data (*Hattne et al., 2014*; *Kern et al., 2012*; *Lyubimov et al., 2016*; *Sauter, 2015*) and therefore pose challenges for data processing. Early XFEL diffraction data sets were processed exclusively using 'Monte Carlo' summation methods (*Kirian et al., 2010*), which required large numbers of diffraction images.

Previously, we described a program, *PRIME*, that uses post-refinement techniques to improve the scaling and merging of XFEL data sets obtained from relatively small numbers (100–2000) of diffraction patterns (*Uervirojnangkoorn et al., 2015*). This method, and similar methods described by others (*Kabsch, 2014*; *Kroon-Batenburg et al., 2015*; *White, 2014*; *Ginn et al., 2015a*) were applied to diffraction data of crystals of known structure (*White, 2014*; *Kabsch, 2014*; *Lyubimov et al., 2015*; *Murray et al., 2015*; *Ginn et al., 2015b*). We subsequently successfully applied our methods to the previously unknown crystal structure of the complex between synaptotagmin-1 (Syt1) and the neuronal SNARE complex, which mediates the fusion of synaptic vesicles with the synaptic membrane and is essential for $Ca^{2+}$-dependent neurotransmitter release (*Zhou et al., 2015*). We had only a limited number of relatively large, plate-like crystals available that were not suitable for liquid jet experiments, so the XFEL diffraction data were collected on the goniometer setup implemented at the X-ray Pump Probe (XPP) endstation of the Linac Coherent Lightsource (LCLS) at SLAC National Accelerator Laboratory (*Cohen et al., 2014*).

To date, several structures have been determined using relatively small numbers of diffraction images obtained from crystals of known structure that diffracted to high resolutions (*Cohen et al., 2014*; *Lyubimov et al., 2015*; *Uervirojnangkoorn et al., 2015*; *Hirata et al., 2014*; *Suga et al., 2015*). Although valuable as test cases for methods development, they were not challenging enough to test the limits of XFEL data processing methods. In contrast, the Syt1–SNARE XFEL diffraction data set contained two crystal forms indistinguishable by visual inspection and had a limiting resolution of ~3.5 Å. These diffraction data required us to improve our data processing methods.

Here we describe improvements to *PRIME* (*Uervirojnangkoorn et al., 2015*) and other parts of the data processing system. We reprocessed the XFEL diffraction data of the Syt1–SNARE complex, which resulted in small improvements to the data and the atomic model refined against these data. We verified the accuracy of these improved methods by processing a simulated a XFEL diffraction data set that mimicked the Syt1–SNARE XFEL experiment. We also compared the reprocessed XFEL diffraction data set to a synchrotron diffraction data set collected from a similar Syt1–SNARE crystal. The synchrotron data extended to lower resolution (4.1 Å) and consequently provided less detailed electron density maps. Nonetheless, comparison with XFEL-derived maps calculated to 4.1 Å resolution showed that the XFEL maps were slightly more interpretable. We conclude that our methods have general applicability to XFEL diffraction data processing.

## Results and discussion

### Reprocessing the 3.5 Å XFEL diffraction data of the Syt1–SNARE complex

As previously described (*Zhou et al., 2015*), we used the program *cctbx.xfel* (*Hattne et al., 2014*) to index and integrate the observed XFEL diffraction images of crystals of the Syt1-SNARE complex. We performed a grid search of spot-finding parameters on an image-to-image basis to maximize the success of indexing and integration (*Lyubimov et al., 2016*). We divided the diffraction images into individual clusters based on their crystal symmetry and unit cell parameters using hierarchical clustering (*Andrews and Bernstein, 2014*; *Zeldin et al., 2015*). Using the largest cluster, we employed post-refinement as implemented in the program *PRIME* (*Uervirojnangkoorn et al., 2015*) to generate a merged diffraction data set from the relatively limited number of diffraction images. Two previously unpublished features were necessary to obtain the best results possible at the time, and are described in detail in the methods. First, the crystal lattice model refinement algorithm in *cctbx.xfel* was enhanced in order to minimize instances of mis-indexing. Second, an improved scaling procedure was implemented in *PRIME*.

Subsequent to the original publication (*Zhou et al., 2015*), we further enhanced the data processing methods. We combined the *IOTA* grid search method (*Lyubimov et al., 2016*) with new features

including automatic rejection of images that had no discernible diffraction, the ability to use information about the Bravais lattice and unit cell dimensions from other data, and detection of mis-indexed images. We also implemented a graphical user interface for the processing of XFEL diffraction images. As described in the Materials and methods, the *cctbx.xfel* algorithms were also modified to allow refinement of parameters such as detector position and tilt. Finally, we introduced a feature to include reflections with negative intensity measurements in merging and post-refinement with *PRIME.*

We reprocessed the XFEL diffraction data of the Syt1–SNARE complex at 3.5 Å resolution to take advantage of all improvements implemented since the original publication (*Zhou et al., 2015*). Of the 789 diffraction images collected from 148 crystals, 362 images were indexed in the 'long unit cell' crystal form (see Materials and methods); this was the largest of the unit cell 'clusters' determined by the Andrews-Bernstein algorithm (*Andrews and Bernstein, 2014* ; *Zeldin et al., 2015*). Of these, 328 images could be successfully integrated. Of the 328 integrated images, 15 were rejected during post-refinement, and the remaining 313 were merged into the final scaled data set (*Table 1A*).

As in the originally published Syt1–SNARE structure, we observed strong electron density for many side chains (*Figure 1A,B*). Our modified data processing methods resulted in small improvements in the refinement statistics of the Syt1–SNARE structure (*Table 1A*) *vs.* the originally published structure (*Zhou et al., 2015*). Moreover, the reprocessed XFEL diffraction data produced slightly more interpretable electron density maps, which in a few cases allowed better modeling of side chain rotamers that were previously difficult to interpret (*Figure 1—figure supplement 1*). Simulated annealing composite omit maps (*Figure 1—figure supplement 2*) indicated that the electron densities observed in the XFEL-data derived maps are not likely affected by potential model bias.

As a further assessment of the Syt1–SNARE XFEL diffraction data reprocessing, we re-refined the most current atomic model against the original XFEL diffraction data, resulting in $R_{work}$ = 31.1% and $R_{free}$ = 33.6%. Since the original XFEL diffraction data did not include negative intensity measurements, we also re-refined the current model against the reprocessed data set including only reflections with positive measurements. $R_{work}$ and $R_{free}$ were 1.1% and 0.4% lower, respectively, than those for the original XFEL diffraction data (*Table 1D*), indicating that the reprocessed data are more accurate than the original data. Inclusion of negative intensity measurements further lowered $R_{work}$ and $R_{free}$ by 0.8% and 0.3%, respectively, for the reprocessed data (*Table 1A*), indicating that the inclusion of negative intensities results in a somewhat more accurate model, which could be due to improved data completeness, accuracy, or both.

## Accuracy of data processing with simulated XFEL diffraction images

In order to assess if our improved data processing system could accurately process XFEL still data, we generated a simulated XFEL diffraction data set from the atomic coordinates of the Syt1–SNARE complex (Materials and methods) and processed it with the same methods used for the observed XFEL data. This produced a merged dataset with excellent $CC_{1/2}$, $R_{merge}$ and I / σ(I) values (*Table 1C*) and good agreement with structure factors calculated from the Syt1–SNARE structure that were used to generate the simulated XFEL data set [CC = 97.5% (88.4% in the high resolution bin), R = 11.8% (35.1% in the high resolution bin)]. The atomic model of the Syt1–SNARE complex was then re-refined against the simulated XFEL dataset, resulting in low R-values (*Table 1C*) and good agreement with the structure that was used to generate the simulated data set (root-mean-square-difference = 0.11 Å). Moreover, electron density maps computed from the simulated XFEL dataset showed strong features for most side chains (*Figure 1C*). Thus, our data processing system can produce a reasonably accurate merged diffraction data set from simulated XFEL still images. However, the $CC_{1/2}$ of the observed XFEL diffraction data and the R values of the corresponding refined atomic model are inferior to those obtained from the simulated XFEL data set (*Table 1A*). Although this difference might arise from experimental noise, it may also indicate that the simulation does not fully account for certain features of the observed XFEL data.

**Table 1.** Data processing and refinement statistics for the synchrotron, XFEL, and simulated XFEL diffraction data.

| | A. XFEL (SLAC-LCLS) | B. Synchrotron (APS-NECAT) | C. Simulated XFEL (nanoBragg) | D. XFEL - Exclusion of negative intensities | E. XFEL - Exclusion of high resolution reflections | F. XFEL - Exclusion of low resolution reflections | G. Simulated XFEL – Exclusion of negative intensities |
|---|---|---|---|---|---|---|---|
| No. images | 313 | 450 | 432 | 297 | 316 | 304 | 432 |
| Space group | $P2_12_12_1$ | $P2_12_12_1$ | $P2_12_12_1$ | $P2_12_12_1$ | $P2_12_12_1$ | $P2_12_12_1$ | $P2_12_12_1$ |
| Cell dimensions* a, b, c (Å) | 69.5, 171.0, 291.3 | 68.8, 169.7, 286.8 | 69.4, 170.4, 291.0 | 69.4, 170.4, 291.0 | 69.5, 171, 291.4 | 69.6, 171, 291.4 | 69.4, 170.4, 291.0 |
| Resolution† (Å) | 20.0–3.5 (3.56–3.50) | 50.0–4.1 (4.21–4.10) | 20.0–3.5 (3.56–3.50) | 20.0–3.5 (3.56–3.50) | 20.0 – 4.1 (4.17–4.10) | 10.0 – 3.5 (3.56–3.50) | 20.0–3.5 (3.56–3.50) |
| Data cutoff [I / σ(I)] | ~3 | ~3 | ~3 | .5 | ~3 | ~3 | .5 |
| Completeness (%) | 97.8 (89.2) | 98.1 (99.1) | 99.8 (99.1) | 87.8 (58.4) | 95.8 (84.2) | 88.3 (58.0) | 99.1 (95.5) |
| Multiplicity (rotation) ‡ | – | 3.3 (3.4) | – | – | – | – | – |
| Multiplicity (still) ‡ | 6.1 (2.9) | – | 9.5 (6.7) | 4.4 (1.67) | 5.8 (3.1) | 4.3 (1.7) | 8.4 (4.8) |
| **Post-refinement parameters** | | | | | | | |
| Linear scale factor $G_0$ | 2.8 | – | 1.9 | 2.9 | 3.9 | 2 | 1.3 |
| B | 66.8 | – | 39.3 | 73.1 | 71.3 | 59.9 | 30.1 |
| $\gamma_O$ (Å⁻¹) | 0.00024 | – | 0.00027 | 0.00016 | 0.00015 | 0.0014 | 0.00032 |
| $\gamma_e$ (Å⁻¹) | 0.00627 | – | 0.00270 | 0.00483 | 0.00498 | 0.0639 | 0.00225 |
| Average $T_{pr}$ | 169.8 | – | 90.7 | 157.22 | 119.8 | 125.6 | 74.8 |
| Average $T_{xy}$ (mm²) | 7.3 | – | 1.5 | 13.74 | 4.1 | 5.8 | 1.35 |
| $CC_{1/2}$ | 94.3 (34.2) | 99.9 (63.8) | 99.1 (82.0) | 93.6 (43.4) | 94.3 (71.0) | 94.9 (41.7) | 99.4 (80.2) |
| $R_{merge}$(%) (rotation) † | – | 12.1 (78.8) | – | – | – | – | – |
| $R_{merge}$(%) (still) † | 49.4 (79.5) | – | 17.3 (53.0) | 36.8 (33.5) | 38.3 (37.1) | 37.1 (32.8) | 13.0 (31.5) |
| I / σ(I) | 3.6 (0.2) | 7.6 (1.8) | 8.1 (1.6) | 4.7 (1.3) | 6.2 (1.8) | 3.5 (1.3) | 8.4 (2.4) |
| **Structure-refinement parameters** | | | | | | | |
| $R_{work}$ / $R_{free}$ (%) | 29.2/32.9 | 28.8/ 29.5 | 9.6/11.1 | 30.0/33.2 | 28.9/33.5 | 31.5/34.6 | 10.7/12.5 |
| R.m.s. deviations | | | | | | | |
| Bond lengths (Å) | 0.002 | 0.004 | 0.002 | 0.002 | 0.003 | 0.002 | 0.003 |
| Bond angles (°) | 0.5 | 0.7 | 0.4 | 0.5 | 0.6 | 0.5 | 0.5 |
| No. atoms | | | | | | | |
| Protein | 10578 | 10889 | 10578 | 10578 | 10903 | 10578 | 10578 |
| $Ca^{2+}$ | 21 | 15 | 21 | 21 | 21 | 15 | 21 |
| B-factors | | | | | | | |
| Protein | 106 | 155 | 40 | 52 | 103 | 64 | 43 |
| $Ca^{2+}$ | 88 | 149 | 27 | 33 | 77 | 43 | 28 |

*The unit cell parameters displayed for XFEL data sets are the mean values of these parameters after post-refinement.

† Values in parentheses are for the highest resolution shell.

‡ '(rotation)' refers to rotation diffraction data collected at the synchrotron and '(still)' refers to XFEL diffraction data.

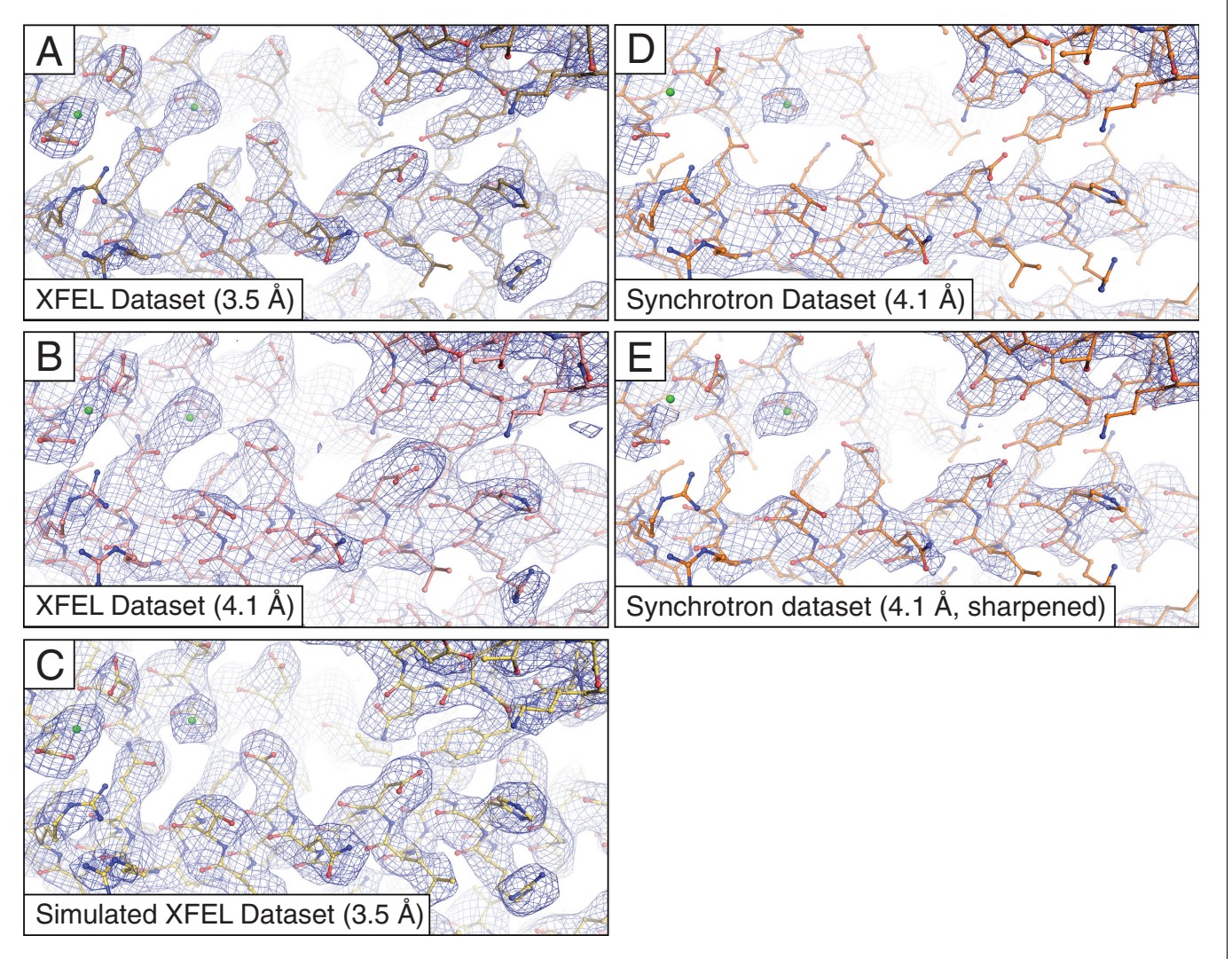

**Figure 1.** Representative $2mF_o$-$DF_c$ electron density maps. The maps were obtained using (**A**) the XFEL diffraction data at 3.5 Å resolution; (**B**) same as (**A**) but truncated to 4.1 Å resolution in order to match the limiting resolution of the synchrotron data; (**C**) the simulated XFEL diffraction data set; (**D**) the synchrotron data set (at 4.1 Å resolution), (**E**) same as (**D**) but with the synchrotron data sharpened with $B_{sharp}$ = −23 Å$^2$ in order to account for the differences in overall B-factors of the corresponding diffraction data sets. All maps were rendered at a contour level of 2.0 σ.

The following figure supplements are available for figure 1:

**Figure supplement 1.** Representative $2mF_o$-$DF_c$ electron density maps of an interface between the C2A and C2B domains of synaptotagmin-1.

**Figure supplement 2.** Simulated annealing composite omit maps generated using $2mF_o$-$DF_c$ coefficients.

## Comparison to 4.1 Å synchrotron diffraction data of the Syt1-SNARE complex

We previously reported diffraction data sets from crystals of the Syt1-SNARE complex collected at a microfocus synchrotron beam line using rotation data collection (*Zhou et al., 2015*). These synchrotron data sets, however, were obtained from a different (short unit cell') crystal form than that for the XFEL-data derived Syt1–SNARE crystal structure (*Zhou et al., 2015*). Here we have measured a synchrotron data set from a similar crystal in the same long unit cell' crystal form used for the XFEL-data derived structure (*Table 1B*).

A key difference between the XFEL and synchrotron data is the higher limiting resolution of the XFEL diffraction data (3.5 Å vs. 4.1 Å, *Table 1A, B*). The synchrotron diffraction data were obtained from a single crystal judged to be the best from a pool screened for optimal diffraction, and 4.1 Å was determined to be the maximum achievable limiting resolution with this particular data set. In contrast, the XFEL diffraction data were obtained from 148 crystals of widely varying quality with limiting resolutions ranging from ∼5 Å to 2.9 Å (in 92 cases). Even though only 313 XFEL diffraction images were used in the final merged data set, it had high completeness (97.8%) and good multiplicity (6.1), along with reasonable merging statistics to 3.5 Å resolution (*Table 1A*). Notably, more diffraction images (450) were used in the synchrotron data set.

For more precise comparison, we re-scaled the synchrotron data to the same isotropic temperature factor (B) value as that of the XFEL data by applying a (sharpening) B factor of $-23$ Å$^2$. Subsequent re-refinement of the atomic model produced only a slight improvement in the electron density map (*Figures 1D,E*). We also tested the effect of the different resolution limits of the XFEL and synchrotron data sets by reprocessing the XFEL data truncated to 4.1 Å resolution, followed by atomic model refinement (*Table 1E*). Although the synchrotron data-derived model refined to a lower $R_{free}$ value than the XFEL data-derived model (29.5% vs. 32.9%), the electron density maps calculated from the XFEL data set (*Figures 1A,B*) generally contained more information than the synchrotron data-derived maps (*Figure 1D*), even when the latter were sharpened (*Figure 1E*). The same effect was found using simulated annealing composite omit maps (*Figure 1—figure supplement 2*), suggesting that the side chain density features of the XFEL data-derived maps are not the result of model bias.

The electron density maps were quantitatively assessed by the real-space correlation coefficient (CC) calculated for each amino acid type (*Figure 2*). The real-space CCs were calculated using *phenix.get_cc_mtz_pdb* (*Adams et al., 2010*) by comparing a likelihood-weighted *2mF$_o$-DF$_c$* electron density map with a map calculated from the model. All amino acid types correlate better with the XFEL data-derived map than with the corresponding synchrotron data-derived map (*Figure 2A*). Additionally, more Ca$^{2+}$ were visible in the XFEL-data derived maps (21 Ca$^{2+}$) than in synchrotron-data derived maps (15 Ca$^{2+}$). The 13 Ca$^{2+}$ that were in matching positions in both maps had higher real-space CCs in the XFEL-data derived structure (*Figure 2C*). Similar, but somewhat less pronounced results were obtained when calculating the real-space correlation coefficients from the simulated annealing composite omit maps (*Figure 2—figure supplement 1*).

Although the differences between XFEL and synchrotron data sets may be due to differences in radiation damage sustained by the crystals during X-ray exposure, they may have arisen from batch-to-batch differences in crystal quality, individual crystal-to-crystal variability, differences in data collection strategy, or a combination of all these factors. Further studies will be required to determine whether XFELs can improve upon the diffraction data obtained from synchrotrons.

## Conclusions

Advances to our XFEL diffraction data processing system resulted in somewhat better statistics of a diffraction data set and refined atomic model for the crystal structure of the Syt1–SNARE complex than that previously reported (*Zhou et al., 2015*). Compared with a lower resolution synchrotron diffraction data set obtained from similar crystals in the same crystal form, the electron density maps calculated from the XFEL data contained more information, especially for many side chains. However, the statistics of the XFEL diffraction data (CC$_{1/2}$) and refined atomic model ($R_{work}$ and $R_{free}$) are still inferior to those obtained from synchrotron data (*Table 1*). The accuracy of a merged data set obtained from simulated XFEL diffraction images (*Table 1*) and the accuracy of an atomic model that was refined against it indicate that the differences in refinement statistics cannot be explained by an inability to adequately recover partiality and scaling information from a 'perfect' XFEL diffraction data set. We expect that further improvements in modeling the properties of XFEL diffraction data (*Hattne et al., 2014*; *Lyubimov et al., 2016*; *Sauter, 2015*), such as pulse-to-pulse variation in the SASE spectrum (*Emma et al., 2010*; *Bonifacio et al., 1994*), along with different modes of XFEL beam generation (*Amann et al., 2012*), should further improve the statistics of the XFEL diffraction data and ultimately approach those of synchrotron crystallography.

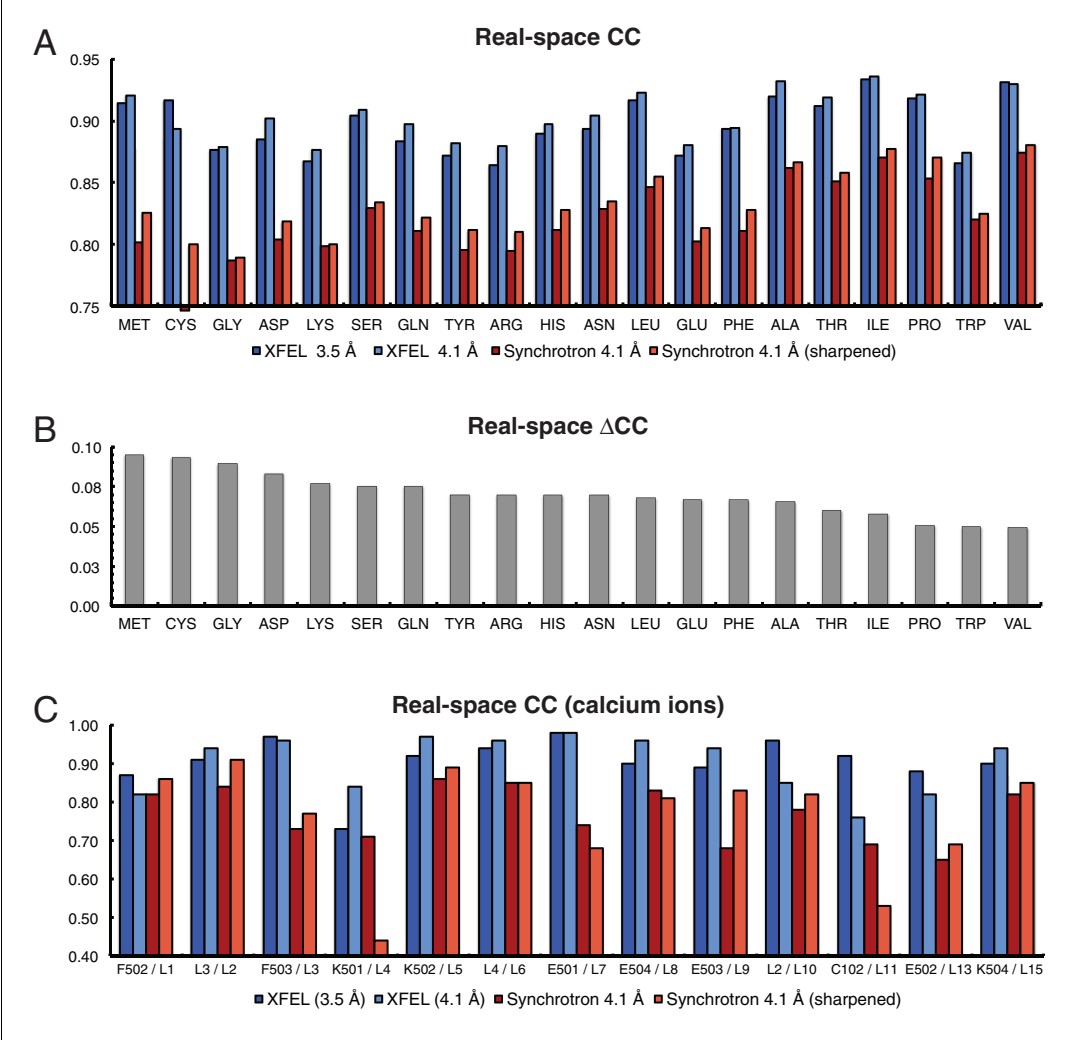

**Figure 2.** Analysis of real-space correlation for atomic models refined against the XFEL and synchrotron data sets. (**A**) Real-space correlation coefficients for atomic models of the Syt1–SNARE complex refined against the XFEL (XFEL 3.5 Å, XFEL 4.1 Å) and synchrotron [Synchrotron 4.1 Å, Synchrotron 4.1 Å (sharpened)] diffraction data and analyzed by amino acid residue type; (**B**) differences between real-space correlation coefficients of the atomic models (ΔCC) refined against the XFEL and synchrotron diffraction data of Syt1–SNARE complex, both processed and refined at 4.1 Å resolution; (**C**) real-space correlation coefficients for the $Ca^{2+}$ sites that were visible in the XFEL- and synchrotron-data derived Syt1–SNARE crystal structures (due to the different numbering of calcium ions in XFEL- and synchrotron-derived structures, two chain and atom labels are given for each, e.g. "F502 / L1"; the first label refers to the XFEL-derived structure, while the second label refers to the synchrotron-derived structure). To facilitate the comparison, the XFEL-based correlation coefficients were calculated at a limiting resolution of 4.1 Å (matching the limiting resolution of the synchrotron data set) as well as at the actual limiting resolution of the XFEL diffraction data set (3.5 Å). Furthermore, the electron density maps obtained from the synchrotron data were sharpened with $B_{sharp} = -23$ Å$^2$ in order to account for the difference in overall B-factors of the diffraction data sets.

The following figure supplement is available for figure 2:

**Figure supplement 1.** Analysis of real-space correlation for atomic models refined against the XFEL and synchrotron data sets versussimulated annealing composite omit maps.

## Materials and methods

### Crystallization of the Syt1–SNARE complex

Construct design, cloning, expression, purification and crystallization of the Syt1–SNARE complex have been previously described (*Zhou et al., 2015*), and are briefly summarized here. Crystallization took 1–3 months, making the optimization of the crystals a difficult and time-consuming process. All

crystals appeared as single plates approximately 25 × 250 × 500 µm$^3$ and were mounted using 0.4–0.7 mm cryo-loops. Due to surface tension, the mounted crystals rested in the same plane as the cryo-loops. The mounted crystals were flash-cooled in a cryo-protecting solution containing the same constituents as the crystallization condition (20 mM Tris-HCl pH 8.0, 300 mM NaCl, 100 mM MgCl$_2$, 1 mM CaCl$_2$, and 0.5 mM TCEP in the protein buffer and 100 mM HEPES-Na pH 7.5 and 1% PEG 8000 in the reservoir buffer) supplemented with 35% (v/v) sucrose. The Syt1–SNARE complex crystallizes in two distinct crystal forms with morphologies that were indistinguishable by inspection of the crystals. As one of these crystal forms arose by the doubling of a single axis of the other crystal form, we term these 'long unit cell' and 'short unit cell' crystal forms, respectively (*Zhou et al., 2015; Zeldin et al., 2015*; *Lyubimov et al., 2016*).

## XFEL data collection

Collection of the Syt1–SNARE XFEL diffraction data has been described (*Zhou et al., 2015*) and is briefly summarized here. The XFEL data were collected at the X-ray Pump Probe (XPP) endstation of the Linac Coherent Light Source (LCLS) at the SLAC National Accelerator Laboratory, using a goniometer-based fixed target sample delivery station and an automatic sample loading system similar to the setup used for conventional synchrotron data collection at SSRL (*Cohen et al., 2002*, *2014*). We used a 30 µm XFEL beam with a pulse duration of 40 fs in SASE mode. We calculated the centroid of the SASE energy spectrum and used this value for the wavelength input to post-refinement of each diffraction image. Each 40 fs XFEL pulse at the XPP endstation at LCLS delivers approximately 10$^{12}$ photons, depositing a dose of 0.5 MGy. A total of 148 crystals were screened, yielding 789 images with usable diffraction.

## Simulated XFEL data

To better understand some of the persistent problems found when integrating the intensities of XFEL data, we simulated XFEL diffraction images (*Table 2*, *Figure 3*) from the previously deposited structure of the Syt1–SNARE complex (PDB ID 5CCG). We calculated structure factors from these coordinates to 3.0 Å resolution with bulk solvent model parameters k_sol = 0.3 e/Å$^3$ and B_sol = 50 Å$^2$ using CNS (*Brunger et al., 1998*). XFEL still diffraction images were simulated using the program *nanoBragg* (http://bl831.als.lbl.gov/~jamesh/nanoBragg/) with parameters shown in *Table 2*. These parameters were optimized using a brute-force grid search scored by the Pearson correlation coefficient between the simulated image and a single observed XFEL diffraction image

**Table 2.** Diffraction parameters for generation of the simulated XFEL diffraction data.

| Parameters | Values |
| --- | --- |
| Beam size (µm) | 30 |
| Spectral dispersion (Δλ/λ,%) | 0.2 |
| Wavelength jitter (%) | 0.5 |
| Intensity jitter* (%) | 100 |
| Beam center X, Y (mm) | 160.53, 182.31 |
| Misset angles (°) | 96.95, −52.09, −32.52 |
| Detector distance (mm) | 299.82 |
| Wavelength (Å) | 1.304735 |
| Mosaicity (°) | 0.2 |
| Divergence (mrad) | 0.02 |
| Dispersion (%) | 0.5 |
| Unit cell dimensions (a, b, c) (Å) | 69.6 171.1 291.9 |
| Mosaic domain size (µm) | 0.96 × 1.0 × 1.1 |

* Intensities were modeled using a Gaussian distribution with mean of 2 × 10$^{12}$ photons/pulse and FWHM of 2 × 10$^{12}$ photons/pulse.

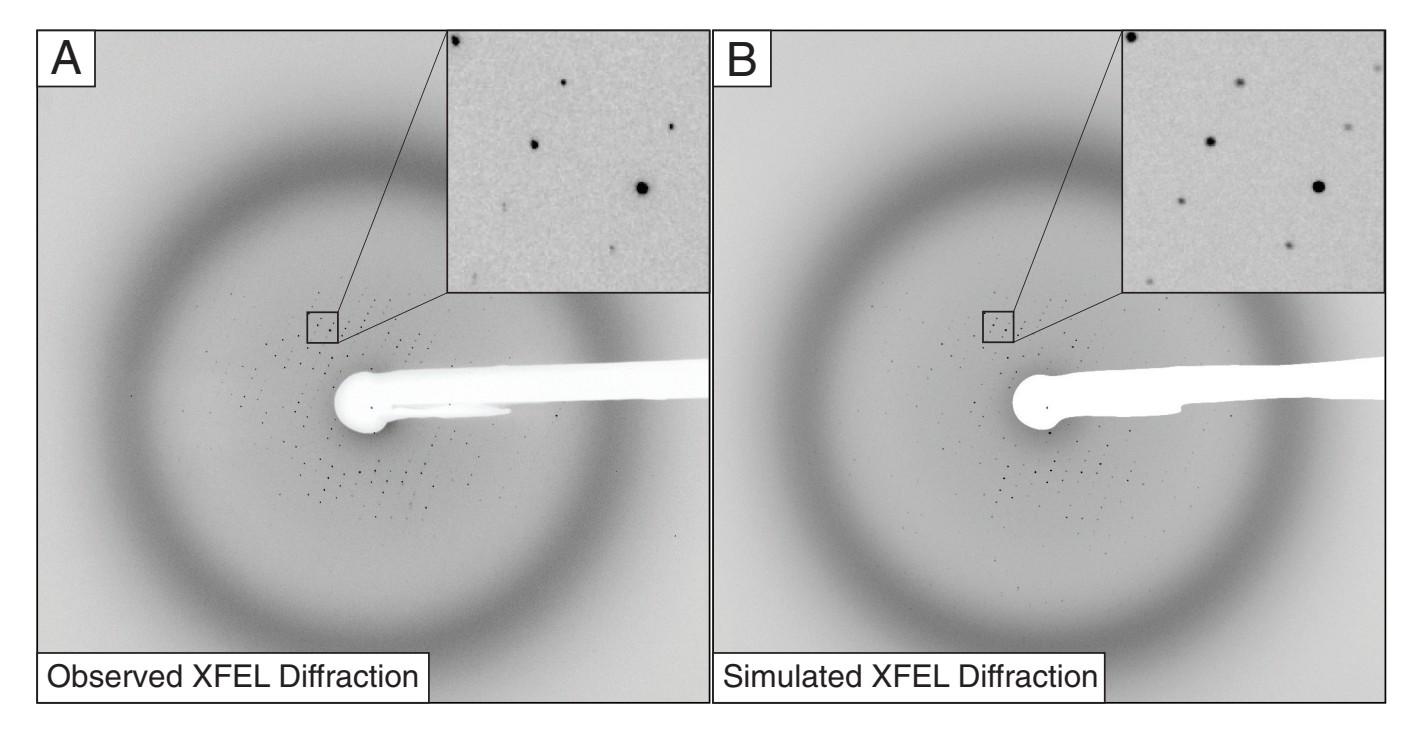

**Figure 3.** Observed (**A**) and simulated (**B**) XFEL diffraction images of the Syt1–SNARE complex. The insets show close-up views of the indicated regions.

with a high number of strong, well-resolved Bragg peaks, hand-selected from the Syt1–SNARE complex XFEL data set. To expedite the comparisons, pixels far from observed spots were excluded using a mask derived from blurring the background-subtracted real image. The point-spread function of the fiber-coupled CCD detector was implemented as described previously (*Holton et al., 2012*), and the conventional mosaic spread (*Helliwell et al., 1982*) was represented by 675 discrete mosaic domains distributed isotropically and randomly over a spherical cap with diameter 0.2 deg. Beam divergence and dispersion were simulated using four discrete source points, separated by the desired divergence, each emitting six discrete wavelengths evenly spaced across the desired spectral width (*Table 2*). The experimentally observed X-ray background was extracted with the *fast-Bragg* companion program *nonBragg* using a median-filtered azimuthal average for constant-resolution rings of pixels. This radial profile was subsequently used to simulate the X-ray background level. Finally, a hand-drawn beam stop shadow was applied to the simulated images. Apart from the random number seed used to generate noise, the final 432 simulated XFEL diffraction images varied from each other only in crystal orientation and the beam intensity, which were randomized shot-to-shot.

## XFEL diffraction data reprocessing

The experimental and the simulated XFEL diffraction images were indexed and integrated using identical procedures (*Figure 4*). We used *cctbx.xfel* (*Hattne et al., 2014*) with the improvements outlined below, along with the latest version of *IOTA* (*Lyubimov et al., 2016*), which enabled proper processing of a few additional diffraction images that had been previously mis-indexed. A hierarchical clustering algorithm (*Andrews and Bernstein, 2014*; *Zeldin et al., 2015*) was used to separate the two crystal forms found in the experimental XFEL diffraction images. Integrated diffraction images were scaled, merged and post-refined using *PRIME* (*Uervirojnangkoorn et al., 2015*) with improvements in scaling as outlined below.

## Improvements to indexing, integration, and scaling

Several previously unpublished improvements to the core modules of the *cctbx.xfel* suite of software were required in order to successfully index and integrate diffraction images obtained from the Syt1–SNARE crystals using XFEL radiation:

1. We found that a small number of images of the Syt1–SNARE XFEL data set were mis-indexed. To alleviate this problem, we added the option to retain the initial assignment of Miller indices to Bragg reflections to *cctbx.xfel* (*Young et al., 2016*). During the indexing step, *cctbx.xfel* determines and refines the three basis vectors that span the primitive triclinic lattice, which are then used to assign Miller indices to the strong Bragg reflections found on the diffraction image. Subsequently, possible crystal symmetry constraints are applied to the lattice model. In the previous work (*Zhou et al., 2015*), Miller indices of the strong spots were then re-deter-mined based on their proximity to nodes on the symmetry-constrained lattice. This was found to be problematic in cases where a long unit cell axis causes lattice nodes to be positioned close together, which may cause incorrect re-assignment of Miller indices. Mis-indexing was suppressed by retaining the original triclinic Miller index assignments throughout the re-refine-ment of the symmetry-constrained lattice model, while applying the appropriate change-of-basis operator to convert the indices to the appropriate symmetry (*Sauter et al., 2006*). Note that this approach does not correct any indexing errors that might have occurred during the initial indexing step.

2. Lattice model refinement is now carried out using modules from the DIALS toolkit (*Waterman et al., 2016*), wherein the target function includes both positions of the observed Bragg reflections and the angular proximity of reciprocal lattice points to the Ewald sphere as described previously (*Sauter et al., 2014*). DIALS allows the refinement of additional parame-ters such as detector tilt and distance, which substantially increases the success of obtaining a lattice model that best correlates with observed diffraction (*Figure 4—figure supplement 1*).

3. Partial reflections with negative intensities after background subtraction are now included in both *cctbx.xfel* and *PRIME*. Approximately 30% of the background subtracted integrated reflection intensities on XFEL diffraction images of the Syt1–SNARE crystals have negative val-ues (*Hattne et al., 2014*; *Sauter et al., 2014*). We found that inclusion of these measurements alleviated unusual behavior of the L-test (*Padilla and Yeates, 2003*) and made the L-test of the XFEL data set comparable to that of the synchrotron data set (*Figure 4—figure supple-ment 2*). (Note that merohedral twinning is not possible in this crystal form.) While further investigations are underway to understand the impact of negative intensities and post-refine-ment on the statistics of the merged data set, we have added an option in *PRIME* to include negative intensity measurements in post-refinement. Currently, the default is to set the thresh-old at $I/\sigma(I) > -3.0$ for reflections used in scale factor and diffraction parameter refinement, and this default was used for the final data set shown in *Table 1A*. The merging statistics for the observed XFEL diffraction data set with negative intensities excluded are shown in *Table 1D*.

## Indexing and integration of the XFEL diffraction images

Typically, when processing XFEL diffraction data of a known system using *cctbx.xfel*, one would sup-ply the known crystal symmetry and unit cell data as target parameters in order to better guide the lattice model refinement. However, this approach is not suitable for a system such as Syt1–SNARE complex, where a batch typically contains crystals in two related, but distinct orthorhombic unit cells (*Zhou et al., 2015*). In this case, using a single target unit cell to process diffraction data from two similar crystal forms would be inappropriate, as incorrect unit cell parameters could be forced upon images. Furthermore, since the Syt1–SNARE complex structure was unknown at the time, and the only information about the unit cell parameters was from lower-resolution synchrotron diffraction data, we were not confident that the available unit cell information would apply to the XFEL diffrac-tion data set. To circumvent these difficulties, we utilized a multi-step data processing strategy (*Figure 4*).

We began by pooling all 789 XFEL diffraction images regardless of which crystal form they were from (*Figure 4A*), and indexed them without supplying any target unit cell parameters (*Figure 4B*). At this stage, we employed a spot-finding parameter grid search using the program *IOTA* (*Lyubimov et al., 2016*), which was specially developed for the purpose of optimizing the process-ing of XFEL diffraction stills. We used the unit cell information obtained from these indexed

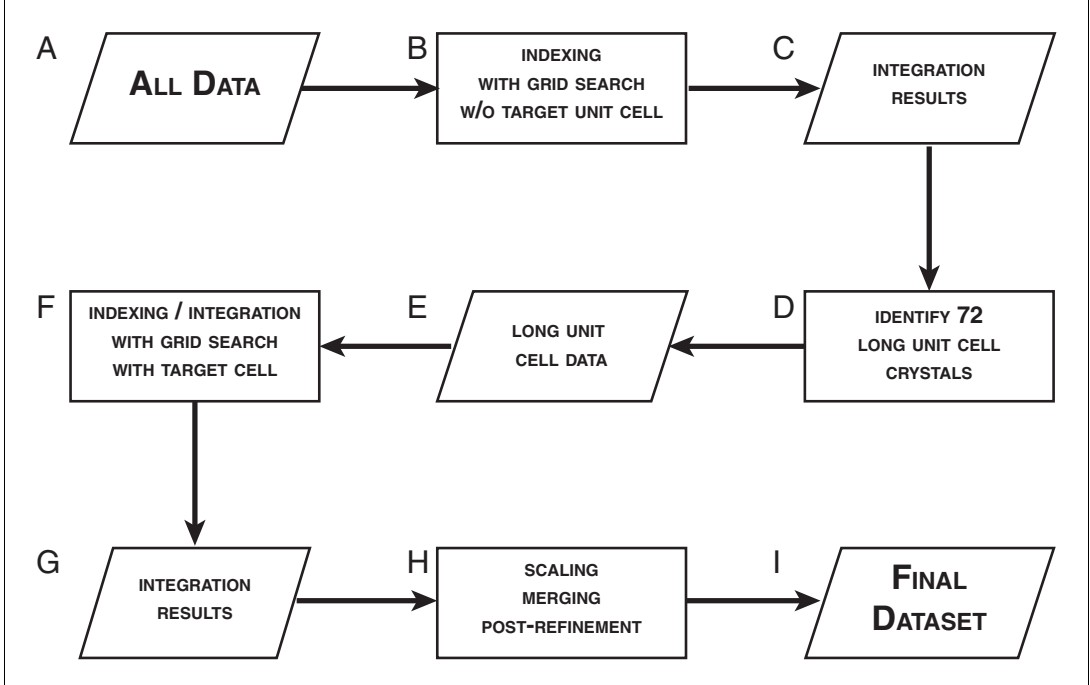

**Figure 4.** Data processing strategy for XFEL diffraction data of the Syt1–SNARE complex.

The following figure supplements are available for figure 4:

**Figure supplement 1.** Improved crystal lattice refinement with DIALS.

**Figure supplement 2.** L-tests for merged diffraction data sets.

**Figure supplement 3.** Distribution of the optimal spot-finding parameter combinations over all successfully integrated diffraction images.

**Figure supplement 4.** Inspection of refined direct beam coordinates indicates possible mis-indexed diffraction images.

diffraction images (*Figure 4C*) and performed a hierarchical cluster analysis of these unit cells (*Andrews and Bernstein, 2014*; *Zeldin et al., 2015*). We then correlated each indexed image with the crystal it was obtained from, and identified 72 crystals that belonged to the 'long unit cell' (a = 69.4 Å, b = 170.8 Å, c = 291.2 Å, $\alpha = \beta = \gamma = 90°$) crystal form (*Figure 4D*). The 362 diffraction images obtained from these 72 crystals comprised the 'long unit cell' cluster (*Figure 4E*). Only this crystal form yielded a sufficient number of diffraction images for a complete data set. The remaining 427 images that were either assigned to the 'short unit cell' cluster, could not be indexed, or contained no interpretable diffraction, were excluded from further analysis.

The clustering algorithm produced a set of consensus unit cell parameters that are assigned to the 'long unit cell' cluster. We used these unit cell parameters as a target for the indexing and integration of the 362 'long unit cell' diffraction images. At this stage, we performed an extensive spot-finding parameter grid search (minimum spot area = 2–22 pixels, minimum spot height = 2–15 $\sigma$, *Figure 4F*). Interestingly, this produced a wide range of spot-finding parameters that would yield optimal integration results (*Figure 4—figure supplement 3A*). An identical grid search carried out for the set of simulated XFEL diffraction images (described below) yielded a much narrower distribution (*Figure 4—figure supplement 3B*), illustrating the high degree of shot-to-shot variability inherent in XFEL diffraction data.

Of the 362 'long unit cell' diffraction images, 328 images were successfully integrated using the wide grid search parameters, while 34 images could not be integrated for a variety of reasons (insufficient number of Bragg reflections, poor diffraction quality, or un-resolvable multiple lattices). The

328 successfully integrated diffraction images were used as input for scaling, post-refinement and merging using the program *PRIME*. Of those 328 integrated diffraction images, 15 were rejected during post-processing due to large deviations from refined mean values for unit cell and scaling parameters. The remaining 313 integrated diffraction images were included in the final merged XFEL data set (*Figure 4G–I*, *Table 1A*).

## Analysis of refined direct beam coordinates identifies mis-indexed XFEL images

Processing of the simulated XFEL diffraction data set with *cctbx.xfel* revealed occasional incidents of mis-indexing by a shift of a Miller index by ± 1. Since even a few mis-indexed frames can adversely affect the statistics of a merged diffraction data set, a diagnostic tool to detect them early would be desirable. We found that a plot of the refined direct beam coordinates can identify mis-indexed diffraction images in the experimental XFEL data set (*Figure 4—figure supplement 4*).

We have shown previously that probable position(s) for the direct beam position on the detector can be deduced from the periodic repeat of bright spots (*Sauter et al., 2004*), given an initial estimate of the beam position derived from the refined detector metrology (*Hattne et al., 2014*). Probability maps for the direct beam position have been useful for data collected at synchrotron beamlines, in cases where the beam position is not correctly recorded with the image metadata. Searching for probable beam positions up to a radius of 4 mm around the initial position allows the indexing program *LABELIT* to estimate the true position. However, we found it counterproductive to apply such a wide beam search to XFEL data. Firstly, at the XPP endstation the beam position with respect to the detector is known within ± 100 μm. Secondly, allowing a large search radius can potentially identify an incorrect beam position. In the case of the Syt1–SNARE complex XFEL diffraction data, eight mis-indexed frames exhibited a shift of ~1.5 mm in beam position, corresponding to a shift of one lattice spacing along the long *c*-axis (291 Å, *Figure 4—figure supplement 4A*). We therefore limited the beam search scope to a radius of 0.5 mm. Under this condition only two XFEL diffraction images remained mis-indexed (*Figure 4—figure supplement 4B*), and were therefore omitted from the merged diffraction data set.

## Scaling, post-refinement, and merging of the XFEL data sets

Integrated XFEL images in the long unit cell cluster were scaled, post-refined and merged with *PRIME,* which corrects partially recorded intensities to their full intensity values using a partiality model (*Uervirojnangkoorn et al., 2015*). This step begins with the generation of the initial reference set, which is in turn used to determine the initial linear scale factor ($G_0$) and the initial temperature factor ($B_0$) for each image. In the original version of PRIME, the initial reference set was obtained by merging the integrated images and scaling them to the mean intensity of these images (referred to as mean-intensity scaling') (*Uervirojnangkoorn et al., 2015*). Our new approach scales each diffraction image to the intensity distribution calculated assuming a random distribution of atoms in the unit cell, *i.e.*, a Wilson plot, generated using the scattering factors of atoms with the temperature (B) factor equal to zero and the contents of the asymmetric unit. For each diffraction image, the full intensity of each reflection is calculated using the initial parameters (crystal orientation, unit cell, mosaicity, spectral dispersion; see [*Uervirojnangkoorn et al., 2015*]). The average of these full intensity estimates is computed for selected reflections (I/σ(I) > 0.5) in equivolume resolution shells to generate an 'observed' Wilson plot, which is fitted to the calculated Wilson plot over the entire resolution range using a linear scale and B factor. Specifically, using the relation

$$\ln \frac{\langle I_{full}(hkl)\rangle}{\sum_i f_i(s)} = \ln G_0 - \frac{B_0 s^2}{2} \tag{1}$$

where $s$ is $2\frac{sin\theta}{\lambda}$, $I_{full}(hkl)$ is the partiality-corrected observed intensity of Miller index *hkl*, and $f_i(s)$ is the scattering factor of atom in each resolution bin, we obtain the initial scale factors ($G_0$ and $B_0$, i.e., the intercept and slope) that optimally fit $\ln \frac{\langle I_{full}(hkl)\rangle}{\sum_i f_i(s)}$ vs. $s$. All integrated diffraction images were brought to the same scale using these initial scale factors, with all reflections included in the merging step. We note that the non-ideal Wilson behavior of macromolecular diffraction data leads to non-

zero values for the B factor of the merged and scaled data set. We refer to this scaling method as 'pseudo-Wilson' scaling.

The next step starts with the pseudo-Wilson scaled and merged data set described earlier as an initial reference and is used to refine crystal orientation, reflection width, and unit-cell parameters. The resolution cutoff was placed where $CC_{1/2}$ fell below 0.25, yielding a merged data set that was 97.4% complete to 3.5 Å resolution. Post-refinement was performed in ten cycles, and at the end of each cycle a new reference set was generated by applying the new scale factors and diffraction parameters to each diffraction image and re-merging the data set. Completeness and average number of observations of the final merged data set and improvement in terms of $CC_{1/2}$ from the starting reference set to the final merged data set are illustrated in *Figure 5*. Wilson plots of the observed intensities before and after mean-intensity scaling and pseudo-Wilson scaling are shown in *Figure 5—figure supplement 1A*. The convergence behavior of post-refinement parameters (scale

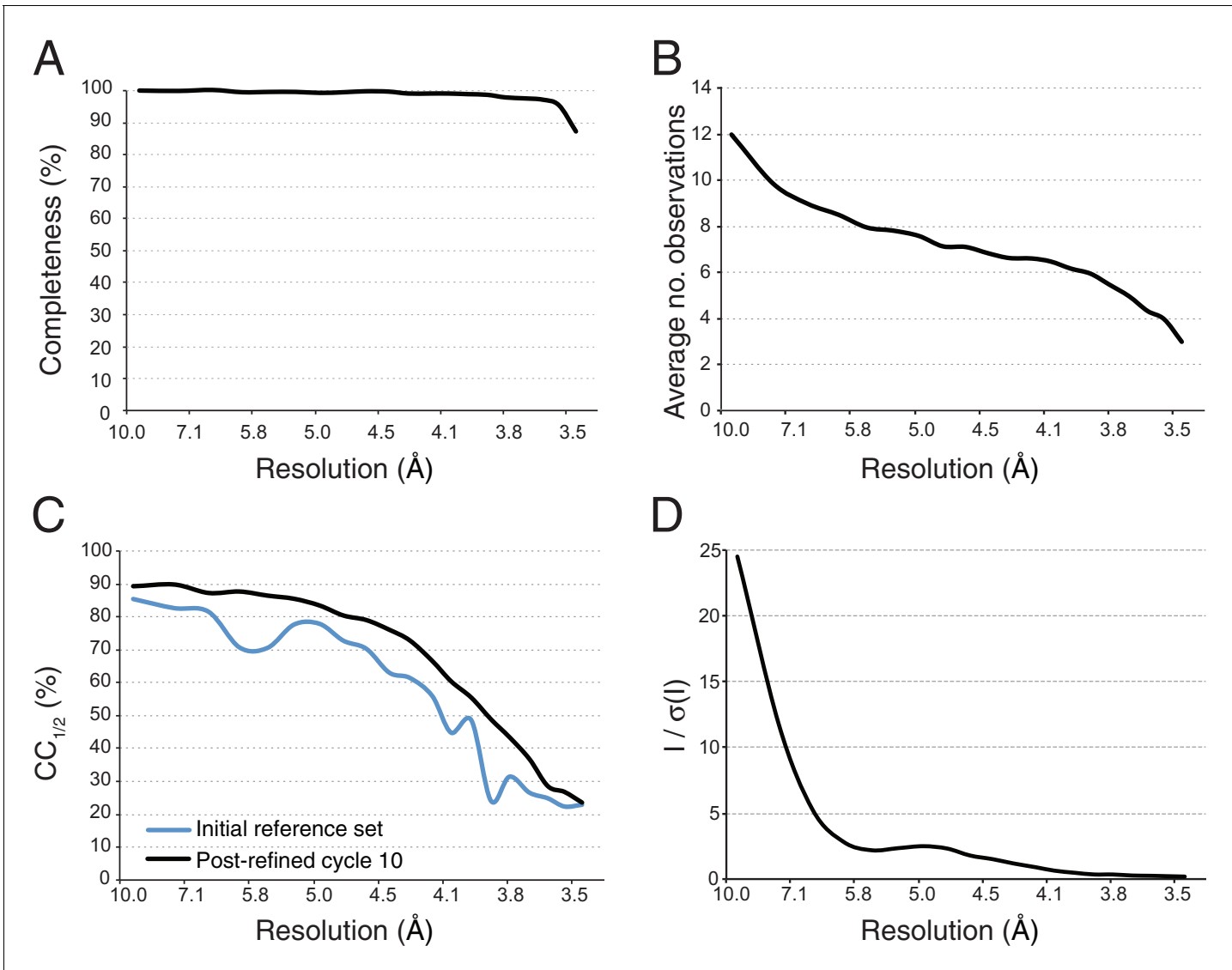

**Figure 5.** Statistical charts for scaled, merged and post-refined XFEL diffraction data. (**A**) Completeness, (**B**) Average number of partial observations per Miller index, (**C**) $CC_{1/2}$, comparing the initial reference set and post-refined data set, (**D**) I / σ(I).

The following figure supplement is available for figure 5:

**Figure supplement 1.** Wilson plots of the diffraction images and convergence of post-refinement after ten cycles for the XFEL diffraction data.

factors $G$ and $B$, reflecting range $\gamma_0$ and $\gamma_e$, crystal orientation, and unit cell dimensions) and of the refinement target functions (post-refinement target $T_{pr}$ and spot-position target $T_{xy}$) are shown in *Figure 5—figure supplements 1B–C*. Although some parameters continued to change after three cycles, little change occurred in $CC_{1/2}$ (*Figure 5—figure supplement 1B*). After scaling each diffraction image using the idealized Wilson model, the B-factor of the initial reference was 29 Å$^2$; the B-factor value changed over the next few cycles until stabilizing at 36 Å$^2$ (*Figure 5—figure supplement 1B*). As mentioned above, we generalized *PRIME* in order to include negative intensity measurements in post-refinement.

Taken together, the improvements in XFEL processing methods resulted in better statistics for the XFEL diffraction data set of the Syt1–SNARE complex (*Table 1*) than previously published (*Zhou et al., 2015*). In particular, the reprocessed data set was is more complete (97.8% vs. 87.6%, respectively) than the previously published XFEL data set, has a higher multiplicity (6.1 vs. 5.0) and a better $CC_{1/2}$ (94.3% vs. 92.7%).

## Effects of resolution and signal-to-noise cutoffs

We investigated the effects of applying different filters to the observed XFEL diffraction data prior to post-refinement and merging on the statistics of the merged data set: (a) merging with negative measurements (including all reflections with $I/\sigma(I) > -3$); (b) merging using all reflections with $I/\sigma(I) > 0.5$ and omitting reflections higher than 4.1 Å Bragg spacings; and (c) merging using all reflections with $I/\sigma(I) > 0.5$ and omitting reflections lower than 10 Å Bragg spacings. Inclusion of the negative intensities slightly improved the merging and refinement statistics (*Table 1*), improved the L-test (*Figure 4—figure supplement 2*), and lowered the overall atomic model R values (*Table 1*). Omitting high- or low-resolution data has a small deleterious effect on the merging statistics ($R_{merge}$ and $CC_{1/2}$) and the corresponding refined atomic models (*Table 1*), suggesting that the intensity measurements are of roughly the same quality throughout different resolutions. The electron density map generated from the XFEL data set that included negative intensities yielded higher average B-factor than that generated from the reflection set without negative intensities (~100 Å$^2$ vs. ~50 Å$^2$ respectively), but this had no substantial effect on the interpretability of the electron density maps, so the data set with negative intensities included was used for the final refinement of the Syt1–SNARE complex (*Table 1*).

In contrast to the substantial effect of excluding negative intensities on the L-test statistics of the observed XFEL data, the effect was very small for the simulated XFEL data (*Figure 4—figure supplement 2*). The effect on merging statistics and R-values of refined atomic models for the simulated XFEL data was similarly small (*Table 1*), likely due to the much smaller fraction of negative intensities in the simulated XFEL data (4.1%) than in the observed XFEL data (15.8%).

## Synchrotron data collection and processing

Synchrotron data were collected using the shutterless, continuous rotation method at the Northeastern Collaborative Access Team (NE-CAT) beamline at the Advanced Photon Source at Argonne National Lab on a Pilatus 6M detector (Dectris). 80 cryo-cooled crystals were screened and the best diffraction data (in the long unit cell form) were merged from three data sets collected at three different positions on a single crystal using consecutive spindle angles. A 30 µm beam was used throughout the experiment. Each of the three data sets contained 150 diffraction images in 0.2° frames and an exposure time of 0.2 s. The diffraction images were indexed and integrated using *XDS* (*Kabsch, 2010*) and scaled and merged using *Scala* (*Evans, 2006*).

## Atomic model refinements

The structure of the Syt1–SNARE complex was refined against the merged XFEL diffraction data set to 3.5 Å resolution (*Table 1*) in a manner similar to that previously described (*Zhou et al., 2015*). Briefly, the phases for the XFEL crystal structure of Syt1-SNARE complex were determined by molecular replacement with Phaser (*McCoy, 2007*) using the rat SNARE complex (PDB ID: 1N7S), the rat Syt1 C2A domain (PDB ID: 3F04), and the rat Syt1 C2B domain (PDB ID: 1UOW) as search models. The structure was iteratively rebuilt and initially refined using CNS v1.3 (*Brunger et al., 1998*), with deformable elastic network (DEN) restraints (*Schroder et al., 2014*), restrained grouped atomic displacement parameters (ADP) and non-crystallographic symmetry (NCS) restraints, then further

refined with *phenix.refine* (*Adams et al., 2010*) using NCS restraints, secondary structure restraints, and individual ADP refinement. The unit cell dimensions for refinement were set to the mean values obtained by post-refinement with *PRIME*. Re-refinement of the Syt1 – SNARE structure against the reprocessed XFEL data resulted in better atomic model $R_{work}$ / $R_{free}$ values (29.2% / 32.9% vs. 32.2% / 35.3%) than the originally published structure. The final refinement cycle was replicated for the resolution- or intensity-truncated XFEL data sets (*Table 1*) in order to obtain comparable refinement statistics.

The phases for the synchrotron diffraction data were determined using molecular replacement and refined in the same manner as above. We transferred the test set of reflections for cross-validation that was used for the refinement against the XFEL data. Refined independently from the XFEL-data derived structure, the synchrotron data-derived structure yielded slightly better $R_{work}$ / $R_{free}$ values than the structure refined against the XFEL diffraction data truncated to the same resolution (4.1 Å) (*Table 1*).

## Generation of simulated annealing composite omit maps

Composite omit maps were generated in order to reduce the potential effect of model bias (*Figure 1—figure supplement 2*). The maps were generated using *phenix.composite_omit_map* (*Terwilliger et al., 2008*) via the Phenix GUI (*Echols et al., 2012*), employing a single cycle of Cartesian simulated annealing (starting at 5000 K) to reduce model bias, followed by minimization. We chose to exclude bulk solvent from the omitted regions, as that option appeared to result in stronger omit map features.

## Additional files

The merged XFEL diffraction data (*Table 1A*) and the merged synchrotron diffraction data (*Table 1B*) for the Syt1–SNARE complex, along with the corresponding atomic model coordinates have been deposited in the Protein Data Bank (PDB IDs 5KJ7 and 5KJ8, respectively). The merged simulated XFEL diffraction data (*Table 1C*) and corresponding atomic model coordinates are available as Source Data files. The complete set of raw XFEL diffraction images for the Syt1–SNARE complex will be deposited in the SBGrid data repository.

## Acknowledgements

The authors would like to thank the National Institutes of Health (R01GM102520 and R01GM117126 to NKS, and NIGMS P41 GM103393 to WIW, which also supports JMH). ATB and WIW acknowledge a Howard Hughes Collaborative Innovation Award. This work is based upon research conducted at the Northeastern Collaborative Access Team beamlines, which are funded by the National Institute of General Medical Sciences from the National Institutes of Health (P41 GM103403). The Pilatus 6M detector on 24-ID-C beam line is funded by a NIH-ORIP HEI grant (S10 RR029205). Furthermore, this research used resources of the Linac Coherent Light Source (LCLS) at SLAC National Accelerator Laboratory, supported by the U.S. Department of Energy, Office of Science, Office of Basic Energy Sciences under contract no. DE-AC02-76SF00515. This research also used resources of the Advanced Photon Source, a US Department of Energy (DOE) Office of Science User Facility operated for the DOE Office of Science by Argonne National Laboratory under Contract No DE-AC02-06CH11357.

## Additional information

### Competing interests

ATB: Reviewing editor for *eLife*. WIW: Reviewing editor for *eLife*. The other authors declare that no competing interests exist.

### Funding

| Funder | Grant reference number | Author |
| --- | --- | --- |
| Howard Hughes Medical Institute | Collaborative Innovation Award | William I Weis<br>Axel T Brunger |

| National Institutes of Health | R01GM102520 | Nicholas K Sauter |
| National Institutes of Health | R01GM117126 | Nicholas K Sauter |
| National Institute of General Medical Sciences | P41 GM103403 | Axel T Brunger |
| National Institutes of Health | S10 RR029205 | Axel T Brunger |

The funders had no role in study design, data collection and interpretation, or the decision to submit the work for publication.

### Author contributions
AYL, MU, JMH, WIW, ATB, Conception and design, Analysis and interpretation of data, Drafting or revising the article; OBZ, Contributed a computer program, Contributed unpublished essential data or reagents; QZ, MZ, Acquisition of data, Analysis and interpretation of data; ASB, TM-C, NKS, Analysis and interpretation of data, Drafting or revising the article

### Author ORCIDs
Axel T Brunger, http://orcid.org/0000-0001-5121-2036

## Additional files

### Major datasets
The following datasets were generated:

| Author(s) | Year | Dataset title | Dataset URL | Database, license, and accessibility information |
| --- | --- | --- | --- | --- |
| Lyubimov AY, Uer-virojnangkoorn M, Zhou Q, Zhao M, Sauter NK, Brewster AS, Weis WI, Brunger AT | 2016 | Raw XFEL diffraction images (MarCCD format) of Synaptotagmin-1 / SNARE complex, associated with PDB entry 5KJ7 | http://dx.doi.org/10.15785/SBGRID/365 | Publicly available at the Structural Biology Data Grid |
| Lyubimov AY, Uer-virojnangkoorn M, Zhou Q, Zhao M, Sauter NK, Brewster AS, Weis WI, Brunger AT | 2016 | Structure of the Ca2+-bound synaptotagmin-1 SNARE complex (long unit cell form) - from XFEL diffraction | http://www.rcsb.org/pdb/explore/explore.do?structureId=5KJ7 | Publicly available at the RCSB Protein Data Bank (accession no: 5KJ7) |
| Lyubimov AY, Uer-virojnangkoorn M, Zhou Q, Zhao M, Sauter NK, Brewster AS, Weis WI, Brunger AT | 2016 | Structure of the Ca2+-bound synaptotagmin-1 SNARE complex (long unit cell form) - from synchrotron diffraction | http://www.rcsb.org/pdb/explore/explore.do?structureId=5KJ8 | Publicly available at the RCSB Protein Data Bank (accession no: 5KJ8) |

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
