## [Decision Letter]

Thank you for submitting your article "Advances in XFEL diffraction data processing applied to the crystal structure of the synaptotagmin-1 / SNARE complex" for consideration by *eLife*. Your article has been favorably evaluated by Arup Chakraborty as the Senior Editor and three reviewers, including Tom Terwilliger (Reviewer #2) and Stephen C. Harrison (Reviewer #1), who is a member of our Board of Reviewing Editors.

The reviewers have discussed the reviews with one another and the Reviewing Editor has drafted this decision to help you prepare a revised submission.

Summary:

This is a Research Advance that indeed reports improvements in the software previously described in *eLife* (Uevirojnankoorn et al., 2015) and that comes to updated conclusions. It thus conforms appropriately to the intent of the Research Advance category and merits publication.

The authors have made some substantial changes to their approach for analyzing the still images obtained from XFEL data. The new approaches allow treatment of reflections with negative intensities and the use of pseudo-Wilson scaling to normalize not just the average intensity of each image but also the fall-off with resolution (as has been done as a matter of course previously for synchrotron data but not for XFEL data). These approaches were applied to previously analyzed data for a synaptotagmin/SNARE complex, and the resulting maps appear to show more detail than those calculated with data treated by the earlier methods. Moreover, density correlations were better for the XFEL maps than for maps from synchrotron data from similar crystals.

Essential revisions:

Attention to the suggestions (a) and (b) under (1) and to one or more of the suggestions (i), (ii), and (iii) under (2)(b) would make the paper stronger; those calculations should be straightforward and take relatively little time. Many of the other points raised here cannot be settled with the current data, but some comments on those points would be valuable.

1) Improvement from reprocessing XFEL data. The data from the reprocessing are more complete (98% vs. 88%), but a large fraction have negative intensities, and excluding the negative intensities reduces the completeness to 88%, the same as for the previous analysis. The CC_1/2_ is marginally better (94% vs. 93%) for the reprocessed data. The free R value for the complex was lower for the reprocessed data (33% vs. 35%). This is the most convincing evidence that the reprocessed data are more accurate, but as the models were not the same, it is not easy to tell if the change is significant.

Suggestions:

a) It would be helpful to show a detailed comparison of the previous XFEL data with the current data. For example, what is the correlation of intensities between previous and current processed data, and what is the I/σ vs. resolution for each, after selecting the data that are in common between the two?

b) It would also be helpful to refine the same model against the previous and current XFEL datasets, only adjusting parameter values, to have a more convincing comparison based on decrease in the free R.

2) Comparison with synchrotron source. The R-factor and other formal comparisons of the model refinement suggest that the model fits the synchrotron dataset more closely, even when allowing for the resolution difference. This conflict between the free R (overall agreement between model and data) and the real-space correlation (average local agreement between model and data) is unusual. Thus, there appear to be two key issues at this point, neither related to whether the software needs even more work. (a) One key issue, which the authors cannot settle with the current data – if they could, this would be a new paper, not an Advance – appears to be: what is the physical explanation for the differences in the data sets? Excluding for now potentially undiscovered aspects of the scattering physics from the intense XFEL beam, one can see at least three reasons why the data might be different: (i) differences in the way data were collected and merged; (ii) differences in crystal damage (including, of course, the possibility that there was very little in the XFEL case); (iii) differences in the crystals themselves on the two occasions, separated substantially in time. (b) The other key issue is: why does an atomic model fit one better than the other? A hypothesis that needs testing is that the 2F_o_-F_c_ map calculated from the XFEL dataset has more model bias because of less accurate data. (The rationale behind this hypothesis is that a dataset with zero I/σ everywhere would effectively look just like the model because the 2(F_o_-F_c_) portion of the 2F_o_-F_c_ map would be random and the F_o_ portion would look just like the model).

a) What is the physical explanation for the differences in the data sets?i) The single-crystal data came from three fixed positions, with ccd recording. Current experience shows that this regime gives less accurate intensities than the continuous motion regime made possible by direct x-ray detectors. The latter strategy yields data with more uniform B-factors (instead of a "sawtooth" dependence of B-factor on frame number) and more favorable statistics. Nonetheless, a single crystal is more likely than many crystals to have essentially identical unit-cell contents, packing, etc., from one place to another than will a large number of crystals, and perhaps it is not surprising that despite the greater "blur" (higher net B, etc.), one model fits that set better than it does a set merged from many crystals.

ii) It might be a good idea to include the "sawtooth" B-versus-frame number plot for the single crystals, to show how much damage resulted from the total exposure at each location. Did the initial frames show measurable Bragg intensities beyond 4.1 Å? If so, the right comparison of synchrotron and XFEL might have been, for the former case, to collect only as many frames from each position that showed no evident decay, if necessary using more than one crystal to complete the data (but of course getting into an issue of crystal-to-crystal variation – perhaps minimized by taking crystals from a single drop).

iii) With two molecular complexes in the asymmetric unit, there are probably more possibilities for batch-to-batch variation (let alone one year to the next variation) from many small crystals than with just a couple of much larger ones.

b) Why does an atomic model fit one better than the other? Some possible tests for model bias and related issues are suggested below.

i) Calculate a composite omit map (with refinement) instead of a 2F_o_-F_c_ map and analyze this map. For example one could do this by: (1) adjusting the Wilson B of the synchrotron and XFEL datasets to the same value. (j2) removing anisotropy, (3) setting the cell constants of the two datasets to both be equal to the APS dataset, (4) refining the APS model against both datasets, (5) calculating the refined composite omit map, and (6) comparing the map to the model. When one of the reviewers (TT) did this, he found adjusting the cell dimensions didn't change the Rfree for the XFEL dataset very much (R_free_=0.34 compared to 0.33 using original cell parameters). He obtained very similar correlations of XFEL and APS composite omit maps to their respective refined models in the region of the models (0.75 for XFEL vs. 0.74 for APS). The correlation calculated over the entire unit cell was somewhat better for the APS data (0.62 vs. 0.59). This result would suggest that the APS data are at least as accurate as the XFEL data.

ii) Calculate a F_o_(XFEL) – F_o_(synchrotron) map phased with model phases (either model). This again requires the assumption of equal cell dimensions. This map should show differences between the density corresponding to the two datasets and would be expected to have peaks at positions of side chains that are radiation sensitive (or perhaps at sulfur positions). TT calculated such a map and could not see any such pattern of density in the map (it looked almost random but not quite as there were more peaks in the protein region than in the solvent). This did not provide any evidence for a systematic difference between the two datasets at positions of radiation-sensitive atoms. It is possible that the cell dimensions really are different, affecting the difference map (although the similar free R value obtained above with different cell dimensions would argue against this interpretation).

iii) Calculate correlation or R-values of data from APS data collected at lower dose or higher dose with XFEL data. This could be done by processing just early frames of APS data or just late frames of this data and comparing each to the XFEL data. If radiation damage is less in the XFEL data it might be expected to correlate better with the early APS data (though other interpretations would still be possible). See also (a)(ii), above.

The authors suggest that the XFEL dataset had less radiation damage because the correlation difference was bigger for those side chains known to be more sensitive to radiation damage. But among the 7 side chains with the largest differences, four (Gly, Lys, Ser and Gln) are not thought to be sensitive, while Glu, one of the most sensitive, is only 13th in the sorted list. Are the Cys residues involved in di-sulfide bridges? These are far more sensitive than a cysteine. Thus, it seems likely that, while radiation damage may be a contributing factor to the observed differences, it is not the whole story.

In summary, the new software is an advance, but the reviewers are not convinced by the second conclusion – which is in any case not needed to qualify for a Research Advance publication – that "XFELs can improve upon the data obtained from synchrotrons". It has clearly done so (somewhat marginally, but nonetheless properly documented here, including the local real-space correlations, etc.) for this particular synchrotron data set, but the thoughts just outlined illustrate why that data set might not have been optimal and why the comparison may to some extent be apples and oranges.

Thus, the authors should revise the text to avoid conclusions about data "quality" and rather should focus on what this Advance is really about anyway – the new algorithms and software that implements them. To the extent that comparison with synchrotron and simulated data allows them to assess the new methods, inclusion of those comparisons is excellent. But avoid comparing what cannot, at this stage, be properly compared.

---

## [Author Response]

*[…] Essential revisions:*

*Attention to the suggestions (a) and (b) under (1) and to one or more of the suggestions (i), (ii), and (iii) under (2)(b) would make the paper stronger; those calculations should be straightforward and take relatively little time. Many of the other points raised here cannot be settled with the current data, but some comments on those points would be valuable.*

*1) Improvement from reprocessing XFEL data. The data from the reprocessing are more complete (98% vs. 88%), but a large fraction have negative intensities, and excluding the negative intensities reduces the completeness to 88%, the same as for the previous analysis. The CC_1/2_is marginally better (94% vs. 93%) for the reprocessed data. The free R value for the complex was lower for the reprocessed data (33% vs. 35%). This is the most convincing evidence that the reprocessed data are more accurate, but as the models were not the same, it is not easy to tell if the change is significant.*

We agree that the relative improvement due to the reprocessing of the XFEL diffraction data was modest. This was as expected, as the reprocessing was carried out mostly to account for the changes and improvements in the processing software. We believe that the original XFEL-derived structure of Syt1-SNARE was refined with sufficiently good R-values and geometry to merit publication and deposition to the Protein Data Bank.

We would like to clarify that the improvements to the data processing software came in two installments. The first installment (comprising improvements to the software since the publication of articles describing the software) included changes in cctbx.xfel in how the lattice model is generated (specifically concerning the transition from triclinic to higher-symmetry Bravais lattice) as well as the implementation of the pseudo-Wilson scaling in PRIME. These changes were made while the original Syt1-SNARE structure was being refined against XFEL diffraction data. They resulted in the structure published by (Zhou et al., 2015). After the publication of that structure, further changes were made to the software, which resulted in additional improvements to the electron density maps and refined atomic model of the Syt1-SNARE complex described in this Research Advance.

We have added additional details in the Results section of the manuscript, clarifying which changes applied to which version of the Syt1-SNARE XFEL structure. We have also added a new figure supplement to provide an example of the improvements in the atomic model made possible by the most recent methods development (new Figure 1—figure supplement 1).

*Suggestions:*

*a) It would be helpful to show a detailed comparison of the previous XFEL data with the current data. For example, what is the correlation of intensities between previous and current processed data, and what is the I/σ vs. resolution for each, after selecting the data that are in common between the two?*

Comparison of the originally-processed (“old”) and re-processed (“new”) XFEL datasets using the reflections in common shows a significant drop in correlation after ~4.3 Å (Figure 6). These results suggest that the improved XFEL diffraction data processing techniques reported here most strongly affected the high-resolution diffraction data.

Author response image 1.Statistical analysis of XFEL diffraction dataset reprocessing.Correlation coefficients (red line) and R-values (blue line) between original and reprocessed XFEL diffraction datasets.**DOI:**
http://dx.doi.org/10.7554/eLife.18740.017

*b) It would also be helpful to refine the same model against the previous and current XFEL datasets, only adjusting parameter values, to have a more convincing comparison based on decrease in the free R.*

We performed bulk solvent and anisotropic scaling refinement of the new XFEL Syt1-SNARE structure reported here (PDB ID: 5KJ7) against the originally published XFEL dataset resulting in R_work_ = 31.1% and R_free_ = 33.6%. When only reflections with positive measurements were included in the reprocessed XFEL diffraction dataset, both R_work_ and R_free_ were 1.1% and 0.4% lower, respectively, than those for the original XFEL diffraction data. Inclusion of negative intensity measurements further lowered the overall R_work_ and R_free_ values by 0.8% and 0.3%, respectively, for the reprocessed data. These observations indicate that the reprocessing produces data that give somewhat more accurate models. The improved completeness provided by including negative data may also help in this regard. We have added this point to the text.

*2) Comparison with synchrotron source. The R-factor and other formal comparisons of the model refinement suggest that the model fits the synchrotron dataset more closely, even when allowing for the resolution difference. This conflict between the free R (overall agreement between model and data) and the real-space correlation (average local agreement between model and data) is unusual. Thus, there appear to be two key issues at this point, neither related to whether the software needs even more work. (a) One key issue, which the authors cannot settle with the current data – if they could, this would be a new paper, not an Advance – appears to be: what is the physical explanation for the differences in the data sets? Excluding for now potentially undiscovered aspects of the scattering physics from the intense XFEL beam, one can see at least three reasons why the data might be different: (i) differences in the way data were collected and merged; (ii) differences in crystal damage (including, of course, the possibility that there was very little in the XFEL case); (iii) differences in the crystals themselves on the two occasions, separated substantially in time. (b) The other key issue is: why does an atomic model fit one better than the other? A hypothesis that needs testing is that the 2F_o_-F_c_ map calculated from the XFEL dataset has more model bias because of less accurate data. (The rationale behind this hypothesis is that a dataset with zero I/σ everywhere would effectively look just like the model because the 2(F_o_-F_c_) portion of the 2F_o_-F_c_ map would be random and the F_o_ portion would look just like the model).*

Please note that the electron density maps presented in Figure 1 are σ_A_ weighted. This technique effectively down weights the weak data in high resolution shells when generating the maps and minimizes potential model bias. Moreover, as we now document more carefully in the revised manuscript (new Figure 1—figure supplement 1), the 2mF_o_-DF_c_ electron density maps derived from the XFEL diffraction data reveal features not seen in the synchrotron-data derived maps. For example, some side chain rotamers had to be adjusted (or, in some cases, modeled for the first time) in the XFEL-data derived maps. Thus, we conclude that the features revealed by the XFEL-data derived electron density maps are not likely the result of model bias. The composite omit maps calculated with simulated annealing to alleviate model bias (new Figure 1—figure supplement 2) support this assertion.

*a) What is the physical explanation for the differences in the data sets?i) The single-crystal data came from three fixed positions, with ccd recording. Current experience shows that this regime gives less accurate intensities than the continuous motion regime made possible by direct x-ray detectors. The latter strategy yields data with more uniform B-factors (instead of a "sawtooth" dependence of B-factor on frame number) and more favorable statistics. Nonetheless, a single crystal is more likely than many crystals to have essentially identical unit-cell contents, packing, etc., from one place to another than will a large number of crystals, and perhaps it is not surprising that despite the greater "blur" (higher net B, etc.), one model fits that set better than it does a set merged from many crystals.*

The synchrotron diffraction data for Syt1-SNARE were collected using a Dectris Pilatus 6M detector in a shutterless, continuous rotation mode with 0.2° frames (we have added this detail to the Methods section of the manuscript). We agree that variation in unit cell parameters, diffraction strength due to pulse intensity and crystal size differences are likely to give rise to differences with the single crystal diffraction data. We have added this detail to the Results and Discussion section of the manuscript.

*ii) It might be a good idea to include the "sawtooth" B-versus-frame number plot for the single crystals, to show how much damage resulted from the total exposure at each location.*

The R_d_ plots, provided by the XDS package as a crude measure of radiation damage (Diederichs, 2006), show a clear effect vs. exposure (Figure 7). This suggests that synchrotron data collection resulted in noticeable radiation damage to the crystal despite the care that was taken to minimize it.

Author response image 2.Analysis of radiation damage effect on synchrotron diffraction data.(**A**–**C**) R_d_ plot vs. Φ angle for diffraction data collected at three separate positions on the same Syt1-SNARE crystal. Colored boxes represent sub-datasets (green – first 1/3 of the dataset, red – mid 1/3 of the dataset, blue – last 1/3 of the dataset). (**D**) R-factors between the full XFEL diffraction dataset and the three synchrotron sub-datasets; (**E**) Correlation coefficients between the two datasets. The colors for (**D**) and (**E**) follow the same scheme as that of (**A**–**C**); the full synchrotron dataset is shown as a black line.**DOI:**
http://dx.doi.org/10.7554/eLife.18740.018

The R_d_ plots also demonstrate why multi-volume diffraction data collection was needed in this case, and suggests that a serial data collection approach – such as one used in our XFEL experiment – might improve upon synchrotron diffraction data.

*Did the initial frames show measurable Bragg intensities beyond 4.1 Å? If so, the right comparison of synchrotron and XFEL might have been, for the former case, to collect only as many frames from each position that showed no evident decay, if necessary using more than one crystal to complete the data (but of course getting into an issue of crystal-to-crystal variation – perhaps minimized by taking crystals from a single drop).*

Visual inspection of the synchrotron Syt1-SNARE diffraction images shows very weak reflections at ~3.8 – 4.0 Å in the first few images. However, such higher resolution reflections are not visible anymore in subsequent images.

Nevertheless, we attempted to merge the synchrotron diffraction data to beyond 4.1 Å, and noticed a sudden, precipitous deterioration of CC_1/2,_ R_meas_ and I / σ(I) in the resolution shells past that threshold. Attempts to match the 3.5 Å resolution of the XFEL data set resulted in even poorer merging statistics, indicating the lack of any interpretable diffraction at that resolution. Therefore, we believe that we cannot extend the resolution of the synchrotron diffraction dataset past 4.1 Å.

*iii) With two molecular complexes in the asymmetric unit, there are probably more possibilities for batch-to-batch variation (let alone one year to the next variation) from many small crystals than with just a couple of much larger ones.*

We agree that this is definitely possible and have added this point to the Results and Discussion portion of the text.

*b) Why does an atomic model fit one better than the other? Some possible tests for model bias and related issues are suggested below.*

*i) Calculate a composite omit map (with refinement) instead of a 2F_o_-F_c_ map and analyze this map. For example one could do this by: (1) adjusting the Wilson B of the synchrotron and XFEL datasets to the same value. (2) removing anisotropy, (3) setting the cell constants of the two datasets to both be equal to the APS dataset, (4) refining the APS model against both datasets, (5) calculating the refined composite omit map, and (6) comparing the map to the model. When one of the reviewers (TT) did this, he found adjusting the cell dimensions didn't change the Rfree for the XFEL dataset very much (R_free_=0.34 compared to 0.33 using original cell parameters). He obtained very similar correlations of XFEL and APS composite omit maps to their respective refined models in the region of the models (0.75 for XFEL vs. 0.74 for APS). The correlation calculated over the entire unit cell was somewhat better for the APS data (0.62 vs. 0.59). This result would suggest that the APS data are at least as accurate as the XFEL data.*

We have calculated two types of composite omit maps: 1) A “refined” (as per definition in the phenix GUI, not involving simulated annealing) composite omit map suggested in item (5) above, though matching each dataset with the model that was refined against it, rather than using the same model for all calculations; 2) A simulated-annealing composite omit map (with bulk solvent excluded from the omitted regions as implemented in phenix). To facilitate comparison, we calculated omit maps for both the XFEL and synchrotron datasets. In addition, we \ calculated such maps for the XFEL dataset truncated to 4.1 Å and for the “sharpened” synchrotron dataset, with the Wilson B-factor matching that of the XFEL dataset.

For the refined composite omit maps, we obtained similar results to the reviewer, in that the global real-space correlation coefficients (CCs) were fairly similar between the various datasets used. However, the real-space CCs per residue type shows a compelling overall pattern: the 3.5 Å XFEL data usually gave the highest CCs, while truncating the resolution to 4.1 Å yielded similar CCs than the synchrotron data set. Sharpening the synchrotron data had little effect. For individual Ca^2+^ ions, the XFEL datasets have yielded higher CCs in more than half of the cases. For the simulated-annealing composite omit maps (new Figure 1—figure supplement 2), CCs per residue group and for Ca^2+^ ions showed more pronounced differences, with the XFEL data yielding higher CCs in virtually all cases. The resolution effect persists across the board (new Figure 2—figure supplement 1). We have added these points to the text. We included the simulated annealing composite omit maps in this manuscript (new Figure 1—figure supplement 2).

*ii) Calculate a F_o_(XFEL) – F_o_(synchrotron) map phased with model phases (either model). This again requires the assumption of equal cell dimensions. This map should show differences between the density corresponding to the two datasets and would be expected to have peaks at positions of side chains that are radiation sensitive (or perhaps at sulfur positions). TT calculated such a map and could not see any such pattern of density in the map (it looked almost random but not quite as there were more peaks in the protein region than in the solvent). This did not provide any evidence for a systematic difference between the two datasets at positions of radiation-sensitive atoms. It is possible that the cell dimensions really are different, affecting the difference map (although the similar free R value obtained above with different cell dimensions would argue against this interpretation).*

We believe that an F_o_-F_o_ analysis would not be very informative due to crystal and unit cell variability and potential scaling issues between the data sets that would introduce large difference features obscuring the differences in side chain densities.

*iii) Calculate correlation or R-values of data from APS data collected at lower dose or higher dose with XFEL data. This could be done by processing just early frames of APS data or just late frames of this data and comparing each to the XFEL data. If radiation damage is less in the XFEL data it might be expected to correlate better with the early APS data (though other interpretations would still be possible). See also (a)(ii), above.*

For this analysis, we used the synchrotron sub-datasets collected for the three diffraction volumes of the chosen crystal; each of these datasets contained 150 diffraction images, which were separated into three subsets (“first 1/3 of the dataset”, “mid 1/3 of the dataset” and “last 1/3 of the dataset”). We then generated merged datasets for each “time point” by merging the 50-frame (10°) subsets from all three diffraction volumes (Figure 7). While this analysis is useful for addressing the reviewer’s suggestion we believe that it does not pertain to the key messages of this Research Advance so have not included it in the revised manuscript.

Calculating the R-values between the XFEL diffraction dataset and the three sub-datasets also reveals a clear dose effect, predominantly beyond ~ 4.6 Å, where the last 50 images have worse R-values vs. XFEL diffraction data than the mid-50 images and far worse R-values than the first 50 images (Figure 7). The plot of the correlation coefficients of the same sub-datasets shows something similar, albeit to a lesser extent (Figure 7).

*The authors suggest that the XFEL dataset had less radiation damage because the correlation difference was bigger for those side chains known to be more sensitive to radiation damage. But among the 7 side chains with the largest differences, four (Gly, Lys, Ser and Gln) are not thought to be sensitive, while Glu, one of the most sensitive, is only 13th in the sorted list. Are the Cys residues involved in di-sulfide bridges? These are far more sensitive than a cysteine. Thus, it seems likely that, while radiation damage may be a contributing factor to the observed differences, it is not the whole story.*

None of the Cys residues are involved in disulfide bridges; furthermore, as steps were taken to minimize radiation damage (specifically, the multi-volume diffraction data collection strategy), we do not expect the effects to be particularly severe. We do agree with the reviewers that radiation damage, while a contributing factor, is not the whole story. We have modified the text accordingly.

*In summary, the new software is an advance, but the reviewers are not convinced by the second conclusion – which is in any case not needed to qualify for a Research Advance publication – that "XFELs can improve upon the data obtained from synchrotrons". It has clearly done so (somewhat marginally, but nonetheless properly documented here, including the local real-space correlations, etc.) for this particular synchrotron data set, but the thoughts just outlined illustrate why that data set might not have been optimal and why the comparison may to some extent be apples and oranges.*

We agree and have revised the text to avoid the appearance of implying that we have conclusively shown that “XFELs can improve upon the data obtained from synchrotrons”. This will be the subject of future investigations.

*Thus, the authors should revise the text to avoid conclusions about data "quality" and rather should focus on what this Advance is really about anyway – the new algorithms and software that implements them. To the extent that comparison with synchrotron and simulated data allows them to assess the new methods, inclusion of those comparisons is excellent. But avoid comparing what cannot, at this stage, be properly compared.*

We thank the referees for a thorough review and have modified the manuscript accordingly.